# The Role of Walking Experience in the Emergence of Gait Harmony in Typically Developing Toddlers

**DOI:** 10.3390/brainsci12020155

**Published:** 2022-01-24

**Authors:** Daniela De Bartolo, Coen S. Zandvoort, Marije Goudriaan, Jennifer N. Kerkman, Marco Iosa, Nadia Dominici

**Affiliations:** 1Clinical Laboratory of Experimental Neurorehabilitation, Santa Lucia Foundation, 00179 Rome, Italy; daniela.debartolo@uniroma1.it; 2Department of Psychology, Sapienza University of Rome, 00185 Rome, Italy; 3Department of Human Movement Sciences, Amsterdam Movement Sciences, Institute Brain and Behavior Amsterdam, Vrije Universiteit Amsterdam, 1081 HV Amsterdam, The Netherlands; c.s.zandvoort@vu.nl (C.S.Z.); m.goudriaan@vu.nl (M.G.); jnkerkman@gmail.com (J.N.K.); n.dominici@vu.nl (N.D.)

**Keywords:** gait ratio, motor learning, golden ratio, harmony, walking development

## Abstract

The ability to walk without support usually develops in the first year of a typically developing toddler’s life and matures further in the following years. Mature walking is characterized by the correct timing of the different gait phases that make up a full gait cycle formed by stance, swing, and double support phases. The harmony of mature walking is given by a specific self-similar structure of the ratios between the durations of these phases (stride/stance, stance/swing, swing/double support), that in adults all converge to the golden ratio (phi, about 1.618). The aim of this longitudinal, prospective, experimental study was to investigate the evolution of this gait harmonic structure in the transition from supported to independent walking. In total, 27 children were recruited and recorded at various stages of locomotor development, ranging from neonatal stepping up to an independent walking experience of about six months. Differently from walking speed that progressively increased with age, the gait phase ratios started to converge to golden value only after the first independent steps, suggesting a relation to the maturation of the walking experience. The independent walking experience seems to represent a trigger for the evolution of a locomotor harmonic pattern in toddlers learning to walk.

## 1. Introduction

The acquisition of independent walking represents one of the most important achievements for a child’s development and is characterized by the milestone of the first independent steps.

Differently from other mammals, human babies need from 9 to 18 months to master sophisticated coordination, such as locomotion [1,2]. In addition, newly walking toddlers fail to implement the classic pendulum behavior, but it develops over the first few months of independent walking along with other gait parameters [2,3,4,5].

Researchers have aimed to understand how toddlers learn to walk by viewing the onset of independent locomotion as a crucial point [3,6,7], trying to identify the factors underlying the sequences of stages occurring throughout the maturational period of independent locomotion [8,9,10]. In particular, neuromuscular primitives have been investigated in neonates’ stepping, suggesting that they may represent locomotor precursors for mature locomotion [11,12]. This approach has generated the evidence that links the development of locomotion to the progressive neural maturation. Therefore, each one of these stages could reflect the current state of neuromuscular development of the infant [13].

However, it is necessary to also recognize the fundamental role of the interaction with the environment, motor experience, and the capacity of managing upright stability counteracting gravity force in the development of independent walking [14,15,16]. Many different factors are responsible for physical development, which in turn allow alternated, synchronized, symmetric, and rhythmic movements [17]. The literature agrees that in adults, who walk at a comfortable speed, foot off reliably occurs at 60 to 62% of a gait cycle [18,19]. During typical development, stance phase is directly correlated with walking speed but not significantly dependent on the age of the children [20]. Moreover, in different walking conditions, such as walking on a steeply inclined surface [21], the foot-off timing undergoes a low variability, suggesting the possibility that the proportion between stance and swing (60–62% versus 40–38%) is largely unaffected by external conditions [20,21,22]. It has been noted that this proportion is close to the irrational numbers phi (φ), that is 1.6180… [23]. This number characterizes many biological self-similar structures in which the same proportion was found at different levels, such as in fractal geometry [24]. This proportion derives by the ancient geometrical problem, cited by Euclid, to cut a segment so that the length of the whole segment stands to that of the longer part, as well as that of longer part stands to that of the shorter one [24]. The problem is solved by a cut dividing the segment in a proportion equal to 61.8–38.2%, that is very close to the division in stance and swing phases of the physiological gait cycle. It implies that the proportion between the duration of stride and that of stance, as well as that of stance and swing, are both equal to phi [23]. Given the property of autosimilarity of this proportion, the ratio between the shorter part and the difference between longer and shorter parts is equal to phi as well. In the hypothesis of symmetric walking, the difference between stance and swing phase coincides with the sum of the durations of the two double support phases, implying that also the ratio between the duration of the swing phase and the total time spent with both the feet in contact with the ground coincides with the golden ratio [23]. These autosimilarities between the three ratios (stride/stance, stance/swing, and swing/double support phases) provides a harmonic structure to the gait cycle (Appendix A) [23,24].

Furthermore, for body anthropometry, the golden ratio has been found to approximate the proportion of the lengths of consecutive body segments well [24]. The presence of the golden ratio in body proportions has suggested the possibility that it is transferred to gait phases because of the pendular mechanism of walking. The pendular mechanism of walking was a model proposed by Cavagna and colleagues [25] assuming that the body rotates as an inverted pendulum over the foot in contact with the ground during walking. For the isochronism law of pendulum, the period of oscillations was related to the length of the pendulum and not to its mass. To test if the golden ratio is related to this mechanism, whether an artificial alteration of body segment length could modify the ratio between stance and swing was tested, whereby a positive answer was found [26]. A counterproof was given by the fact that this proportion was not altered by an artificial alteration of the weights of body segments [26]. Furthermore, according to the pendulum model, during walking there is a phase in which potential energy is transformed in kinetic energy and another one in which the opposite transformation occurs [25]. However, with respect to an ideal pendulum, gait needs muscular activations synchronized with the limb oscillations for compensating the energy loss. Serrao and colleagues observed that the energy expenditure and step-by-step gait variability have been minimized when the ratio between stance and swing coincides with the golden ratio, confirming that this proportion could be a condition for optimizing pendular walking [27].

It has been shown that pendulum-like walking matures within a few months of independent walking experience [4], while preserving some primitive aspects of neonates’ stepping reflex [11]. In one study, where adults were asked to step in place, showed a proportion between stance and swing of 1.66, significantly different from the golden ratio [22], but no studies have investigated if this ratio during the stepping reflex of newborns is closer to the golden ratio. Furthermore, no studies have investigated if the autosimilar structure of the gait cycle is developed already before the first independent steps, if it is progressively developed with age, or if it is triggered by the first steps. To investigate the above alternative hypotheses, we measured the steps of neonates and toddlers before and after the development of independent walking in this exploratory study.

## 2. Materials and Methods

This longitudinal, prospective, experimental (but not interventional) study was conducted in full compliance with the Declaration of Helsinki for research involving humans. The local ethical committee of the Faculty of Behavioral and Movement Sciences Amsterdam (study protocol No. VCWE-2016-082) approved the experimental procedures. The responsible investigators informed the children’s parents about the study procedures prior to inclusion into the study and asked to provide written informed consent. For all measurements, at least one parent was present during the experiments together with the responsible researchers. Participants visited the BabyGaitLab laboratory of the Department of Human Movement Sciences at the Vrije Universiteit in Amsterdam, wore a diaper and walked without shoes. The specific laboratory settings and experimental procedures were adapted to children so that any risk was equal or lower to that of walking at home. Families were reimbursed for travel expenses and children received a small toy as thanks for their participation.

### 2.1. Participants and Sessions

By word of mouth, we enrolled 27 children (12 females and 15 males, all born at term) with typical development (TD). Exclusion criteria were the presence of known neurological and developmental diseases. To assess the emergence of the independent steps, regular contact was established with the parents to monitor their child’s walking ability. Recording sessions of their first independent steps were scheduled when the parents reported that the child was able to walk independently for at least four consecutive steps. This moment was defined as “walking onset” with which we determined the corresponding “walking age”, and estimated time since the onset of independent walking (equal to age minus age at first steps, this parameter is negative for sessions recorded before the first steps).

Children were evaluated from one up to seven times (sessions) along a period of approximately two years. For our enrolled 27 children, we recorded a total of 110 sessions. To analyze the development of walking within a subject, we also performed a longitudinal analysis on a sub-group of 13 children who had a number of experimental sessions going from four up to seven covering the different phases of the locomotor development, temporally divided by age and walking ability achieved by each participant (this subdivision is defined by Group 1–Group 7, which correspond to each of the subgroups). The first four groups of sessions were the ones before the first independent steps emerge, during these sessions children were supported by the trunk or the arms during walking: between 0–3 months of age—neonate stepping—(Group 1, seven participants: mean: 1.6 ± 0.4 months old), between 3–7 months old (Group 2, nine participants: mean: 5.5 ± 0.9 months old), between 7–10 months old (Group 3, 9 participants: mean: 9.2 ± 1.1 months old), and between 10 months and immediately before the first independent steps (Group 4, 10 participants: mean: 11.2 ± 1.7 months old). Then the first independent steps, sessions were recorded within 2–3 weeks of unsupported walking experience (Group 5, nine participants, mean: 13.1 ± 2.1 months old). Finally, the two groups after the independent walking, when the toddlers were able to walk unsupported: between 1–4 months of independent walking experience (Group 6, six participants, mean: 14.4 ± 1.6 months old), and between 4–7 months of independent walking experience (Group 7, 13 participants, mean: 19.0 ± 1.9 months old).

### 2.2. Procedure

Each child was given time to acclimate to the laboratory and familiarize with the researchers to make the gait recording as ecological as possible. The experimental procedure was adapted to the children such that one researcher and one parent were located next to the child to reduce the risks of falling and to make sure that the child felt comfortable at all times. Children were recorded during treadmill and overground walking. Overground trials were recorded at a lab space with the dimensions of ~5.5 × 3.5 m. Treadmill trials were recorded using a pediatric treadmill, specifically designed for subjects of lower ages (N-Mill 60 × 150 cm, Motek Medical B.V., Amsterdam, The Netherlands). Treadmill speed was tuned to elicit stepping movements and adjusted to a comfortable speed for the child based on his/her walking capacity. Short walking trials (<2 min) were recorded with rest breaks in between.

### 2.3. Walking Conditions

For the recordings of the neonate stepping, a researcher held the infant under their arms with their feet touching the horizontal flat surface of a running treadmill or walkway. Stepping was typically successful when the infants were not drowsy. The neonates were allowed to support as much of their own weight as possible, the rest being supported by the researcher holding the neonate [11,28]. The mean walking speed in neonates was 0.3 ± 0.1 km/h. For the recordings of the supported walking in toddlers, one parent or a researcher held the toddler under their arms or hold both hands/one hand, depending on the walking ability of the toddlers. The mean walking speed during unsupported walking in toddlers was 0.7 ± 0.3 km/h. For the recordings of the first independent steps, one parent initially held the child by hand. Then, the parent started to move forward, leaving the child’s hand and encouraging her/him to walk unsupported on the floor. Also, they were encouraged to look straightforward and to walk as naturally as possible. The mean walking speed during the first steps in toddlers was 1.3 ± 0.5 km/h. For the recordings of the independent walking, one parent or a researcher encouraged the toddler to walk unsupported while playing. The mean walking speed during the independent walking was 2.6 ± 0.5 km/h.

### 2.4. Data Recording

Kinematic and video data were collected with a Vicon system (Vidcon, Oxford, UK) with a sampling rate of 100 Hz. The system used one Vue Vicon camera, place in the sagittal plane, and 10 infrared Vero 2.2 cameras, place around the recording volume. We collected the 3D position of 23 markers placed bilaterally on the skin of the children by a double-sided tape all around the child’s body. For the current study, we specifically focus on the markers attached to the bilateral lateral malleoli and fifth metatarsal–phalangeal joint. The sampling of video and kinematic data was synchronized online. At the beginning of each recording session, anthropometric measurements were taken on each subject. These included the mass, stature of the subject, and lengths and circumferences of the body segments.

### 2.5. Gait Analysis

Foot-contact and foot-off events were manually detected for both sides by visual inspection using a digital video recordings and marker trajectories with Vicon Nexus software (Vicon, Oxford, UK). The gait events were assessed by one experienced researcher while simultaneously monitoring the foot markers’ kinematic data. In particular, the minimum vertical height of the lateral malleoli or fifth metatarsal–phalangeal joint marker (depending on which part of the foot firstly touched the ground) was used to validate the foot-contact events and the increase in the vertical direction of the lateral malleoli after the plateau value during the stance phase was used to validate the foot-off events [11,28,29]. The detected events were then checked by a second researcher and if there was disagreement for more than 4 samples (<6%), the two researchers discussed and re-evaluated the specific stride again to find an agreement. Events associated with gait initiation/termination and turning as well as non-continuous walking or strides characterized by the absence of a double support phase (e.g., jumps or running) were excluded from further analysis. We considered a sequence of strides successful if at least three consecutive strides were present. Step events were then processed with a customized software implemented in MATLAB (MathWorks, 2020b, Mathworks, Nattick, MA, USA).

Gait cycle duration was defined as the time interval between two consecutive foot-contact events of the same leg and the stance phase as the time interval between foot-contact and foot-off, whereas the swing phase was the time between foot-off and the successive foot-contact. The spatio-temporal parameters (stance, swing, and double support durations, and stride length) were calculated for each side and then averaged for all the trials, in order to obtain a single value for each participant/session. Walking speed was calculated as the ratio of the corresponding stride length and stride duration. Gait ratio parameters were calculated according to previous studies [26,27] defining GR0 as the value corresponding to the ratio between the duration of the gait cycle and stance phase, GR1 as that between the stance phase and swing phase durations, and GR2 as between swing phase and double support phase durations. We also computed the normalized speeds through the Froude number (Fr) as Fr = v^2^/(*g*·L), where v denotes the average speed of walking (m/s), *g* represents the gravitational constant (9.81 m/s^2^), and L the leg length as the combined measured of thigh and shank lengths [21]. Normalizing to the walking Froude number is considered suitable when comparing gait patterns at different speeds in participants of different size [4].

### 2.6. Statistical Analysis

Data were assessed by means of (1) a sequential analysis in which data of all sessions were included and (2) a longitudinal analysis in which data were only selected if participants joined more than four consecutive sessions (Group 1 to Group 7).

Due to many different steps being recorded in each trial and to reduce the variability, data were firstly averaged among different steps of each trial (as shown in Appendix B), and then these mean values were averaged among trials.

Descriptive statistics were used to evaluate demographics and average values of gait parameters. A Bisquare Robust Method, Trust-Region Algorithm fit was used, where: f(t) = *a* ∗ exp(−t/τ) + *b* with t being time since onset of unsupported walking (walking age), τ being the time constant, and *a*, *b* as two constants. The fitting was performed to highlight the existence of a trend in the development of the ratio parameters GR, reporting the determination coefficient R^2^ to assess the goodness of the fit. Repeated Measures Analysis of Variance (RM-ANOVA) was used to verify differences among the seven temporal phases of the longitudinal sample. Pearson correlation coefficients were computed to evaluate relations between variables. Levels of significance were chosen at 0.05 for the RM-ANOVA and adjusted to 0.001 for the post hoc comparisons.

## 3. Results

### 3.1. Preliminary Analysis

From the original 110 sessions, we excluded 17 sessions that were characterized by the absence of consecutive steps or by steps lacking a double support phase because performed running or jumping instead of walking. Hence, 93 sessions (reported in Appendix B) were analyzed comprising a group of 27 participants (age range 1.3–23.1 months; individual characteristics are listed in Table A1).

The mean confidence interval of stride to stance ratio among the 93 sessions was 0.06, with a coefficient of variation of 5.9 ± 4.5%. These parameters show the intra-subject variability that resulted lower than the inter-subject variability that could be accounted by the range of variation with respect to the mean value result of 26%.

### 3.2. Fitting Analysis

The scatterplots in Figure 1 illustrate the sequential results of the gait ratio parameters GR0, GR1, and GR2 as a function of age and as a function of walking age for the total database of 93 sequential sessions.

As shown in Figure 1 and Table 1, poor fittings were observed before the first steps for all the three gait ratios, with determination coefficients R^2^ lower than 0.2. In contrast, after first steps, GR data fitted well with the exponential model, with R^2^ ≥ 0.75. Furthermore, constant *b*, the feature of the fitting curve, was very close to the golden ratio, especially for GR0. The time constant τ ranged between 0.2 and 1.4 months.

### 3.3. Longitudinal Analysis

Longitudinal analysis was performed on 13 participants who performed at least four sessions spanning from supported walking until independent walking. Figure 2 reports GR0 for these toddlers computed before the first steps (grey dots), at the first steps (red dots), and after the first independent steps (green dots). When we timelocked the data to the participant’s age (Figure 2A), the variability between participants was considerably high until the age of 15–16 months. In contrast, when we arranged the samples based on walking age (Figure 2B), a specific trend in GR0 values was observed with a decrease in variability immediately after the first steps.

The data of Figure 2B (GR0 vs. walking age) was replicated in Figure 3A superimposed with the mean values (black dots) of data averaged according to the seven groups of children development. The mean values showed an increment of GR0 only after the sessions of first steps and a convergence towards the value of the golden ratio. In Figure 3, this trend was compared with that of walking speed that showed a continuous increment that already started before the first steps.

RM-ANOVA revealed that the differences observed in Figure 3 were statistically significant for GR0 with respect to the time–stage of motor development (F(6,56) = 19, *p* < 0.001). Post hoc analyses showed significant differences for Group 6 and Group 7 with respect to all the other subgroups (*p* ≤ 0.01). GR0 at Group 5 (first steps) did not differ from the stages before the walking onset (Group 5 to Group 1–Group 4: *p* ≥ 0.852).

Walking speed was significantly different across groups (F(6,56) = 64, *p* < 0.001). In particular, walking speed was significantly higher at the first steps (Group 5) compared to most previous stages (vs. Group 1: *p* < 0.001; vs. Group 2: *p* < 0.001; vs. Group 3: *p* = 0.014, but not with that before first steps: Group 5 vs. Group 4: *p* = 0.159). Then, walking speed continued to increase (Group 6 vs. Group 5: *p* = 0.037), but not significant differences were found between Group 6 and Group 7 (*p* = 0.145).

The correlational analysis conducted on the whole longitudinal group revealed a correlation of GR0 with age (R = 0.487, *p* < 0.001) and walking age (R = 0.496, *p* < 0.001), walking speed (R = 0.694, *p* < 0.001), and Froude number (R = 0.678, *p* < 0.001). However, no significant correlations of GR0 with any parameter were found in the four subgroups of toddlers recorded before the start of independent walking. For the three subgroups recorded during the first steps and after the beginning of independent walking, positive and significant correlations were found with respect to age (R = 0.660, *p* < 0.001), walking age (R = 0.697, *p* < 0.001), walking speed (R = 0.852, *p* < 0.001), and the Froude number (R = 0.738, *p* < 0.001). All results obtained from the correlational analysis are summarized in Table 2.

## 4. Discussion

The main purpose of the present study was to investigate if the fractal structure of gait cycle, characterized by a self-similar ratio between gait phases, is already present before the first steps or if it emerges after the onset of independent walking.

Our results showed that a significant trend for gait ratios towards the value of golden ratio started only after the start of independent walking (Figure 1B). The first steps seemed to be a trigger for developing an autosimilar structure of the gait cycle. The fitting parameters confirmed this observation, especially for GR0. In fact, the ratio between stride and stance durations were close to the value of the golden ratio (1.618), achieved with a constant time of only 0.218 months. This result was not obvious; walking speed showed a different trend, with a progressive increment recorded already before the first steps. Differently from walking speed and Froude number, the gait ratio GR0 does not correlate with the walking age before the first steps event, but only after. Therefore, it can be concluded that the walking experience acts as a trigger for the development of a harmonic gait ratio. So, we can state that the acquisition of an autosimilar gait ratio is an event-dependent learning process.

Nevertheless, the autosimilarity of the gait cycle was not the only feature of walking triggered by the first steps. The gait pendular mechanism was found to be developed only after the beginning of the independent walking [4,30]. Harmony of gait cycle and pendular mechanism of walking could be strictly intertwined, indeed both were found associated with the optimization of energy expenditure, to the reduction of gait variability in adult gait [27], while were found to be altered in pathological gait [27,31,32,33].

Usually, before the onset of independent walking, a high variability in locomotor patterns is observed among children and among sessions, but this variability seems to reduce after the first independent steps [5,28,30,33]. In our sample, the gait ratio parameters were chaotic before walking onset, then converging towards the golden ratio with the development of a harmonic walking in less than six months after the first steps. Moreover, this result was not obvious: different from gait ratios, a high variability of walking speed was still observed at Group 7, six months after the onset of independent walking (Figure 3B).

However, studies comparing neonates’ stepping with toddlers and adult walking showed that the two neuromotor primitives extracted by muscular patterns in the steps elicited in neonates are preserved in toddlers’ walking, in which two other new primitives are developed, supporting the idea that this early stepping is a precursor to adult walking [11,12].

In our study, we were able to elicit and record the stepping rhythm in neonates and toddlers before independent steps. However, unlike muscle activation patterns [11,12], the golden gait harmony does not appear before the first steps; despite also being present in neonates, the stance duration was longer than the swing duration (GR1 > 1). It is conceivable that the two locomotor muscular primitives found in newborns were related to the flexion and extension synergies of the stepping reflex, whereas the four primitives of adult walking depict a more complex pattern characterized by four gait phases: the first double support phase, the single leg support, the second double support phase (these three phases are embedded into the stance phase), and the swing phase. Further studies should investigate the relationship of locomotor muscular primitives and the harmonic structure of gait phases. The discovery of the golden section as a physiological value of the gait ratio even in toddlers could have important clinical implications. For example, for an early detection of gait alterations in children with cerebral palsy or other neuromuscular pathologies. The gait ratio could also be a simple and objective measure to assess the effectiveness of neurorehabilitation for these children. Then, if gait harmony can be shaped by walking experience, this could suggest that children may benefit by taking as many steps as possible during motor therapy. Future studies should investigate the role of walking experience in the emergence of harmonic walking in children with neurodevelopmental diseases. Future research should also be conducted to clarify the possible relationships among body lengths, body weight, energy consumption, and walking in the golden ratio. For example, it was found that the mechanical efficiency and work are similar in obese and non-obese children, just the most economical walking speed is reduced in obese children [34]. These results are in line with the previous observation that obese adults rely more on the pendular mechanism rather than on the elastic energy mechanism [35]. Because the pendular mechanism is mainly influenced by body length and theoretically not body segment weights, it could be intriguing to study if the stance to swing ratio is altered in these subjects.

Our study should be read in consideration of its limitations, the most important being the comparison of independent walking with a simulated walking condition in which newborns were recorded on a treadmill and with the support of their body weight. The use of the treadmill as well as the manner in which the babies were supported by the researchers/parents (with trunk support for newborns and with hands support for children before the first independent steps) should be considered as confounding factors. However, previous studies used the same approach to study the walking pattern in infants [4,11,12,28,29]. In fact, several studies used the treadmill in a condition of weight support to elicit stepping in infants [36,37,38]. Furthermore, some patterns recorded during the supported stepping were found to be retained in mature independent walking, such as two out of four locomotor muscular primitives [11]. More recently, it was found that stepping showed stable muscle synergies whose fractionation could account for the synergies of older children who showed a progressive increase of the neuromuscular primitives [12]. These studies suggested that stepping patterns seem to anticipate subsequent developmental changes of locomotion in human babies and they might represent distinct locomotor antecedents [11,12] according to the idea that mature locomotion might stem from neonatal precursor movements [2]. The analysis of stance and swing phases during stepping received less attention, so we applied the same methodology of the above cited studies to investigate the development of gait phases starting from the stepping phases of newborns.

## 5. Conclusions

Many motor abilities developed by children represent skills that evolve progressively, while others are triggered and therefore depend on particular events. In our study, we found that the development of unsupported gait acts as a trigger towards the development of a harmonic gait ratio.

Despite the simplicity with which gait ratio could be compared with the golden ratio to assess the harmony of walking, the development of a harmonic locomotor pattern could be a complex behavior strictly intertwined with other parameters or skills, such as walking speed, balance control [39,40,41,42], and an effective process of perceptual and motor information [43,44]. There is the need to clarify the neural mechanisms underlying the development of walking in the golden ratio, but it seemed evident that the motor interaction with the environment is the prerequisite for an experience-based locomotor learning that guarantees the evolution towards a harmonious locomotor pattern.

## Figures and Tables

**Figure 1 brainsci-12-00155-f001:**
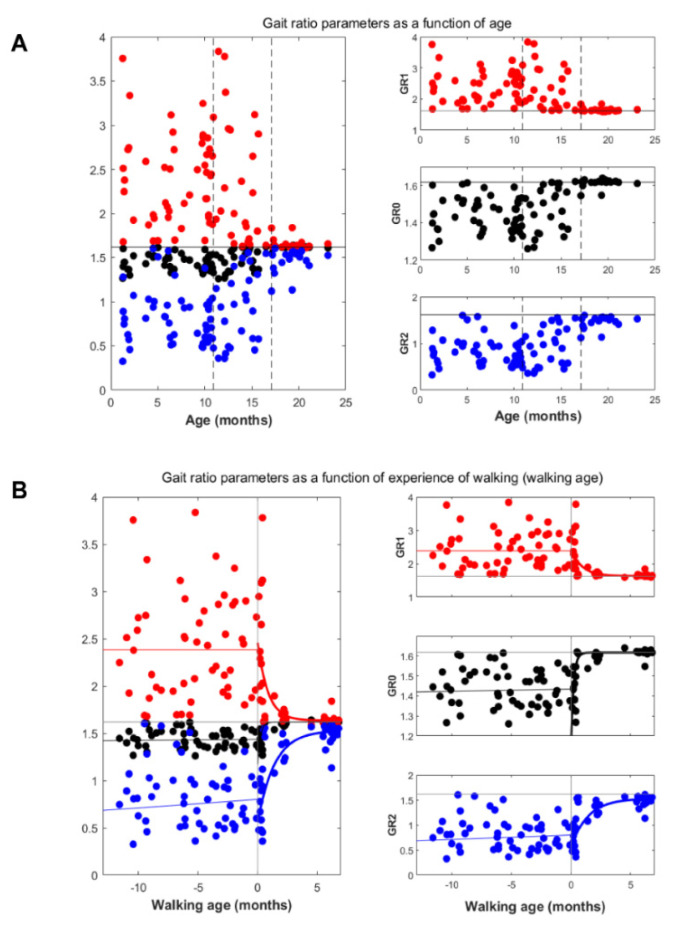
Gait ratios GR as a function of age (panel **A**), and as a function of walking age (panel **B**). In (panel **A**), dashed lines were added to mark the temporal range comprising the minimum and maximal value of walking onset (min and max values: 10.5 and 16.7 months, respectively). The three panels on the right side depict the three parameters: GR0 (black), GR1 (red), and GR2 (blue). The horizontal grey lines mark the value of the golden ratio, which is rounded at 1.618. In (panel **B**), the dashed line marks the beginning of independent walking while a solid curve was obtained by data fitting.

**Figure 2 brainsci-12-00155-f002:**
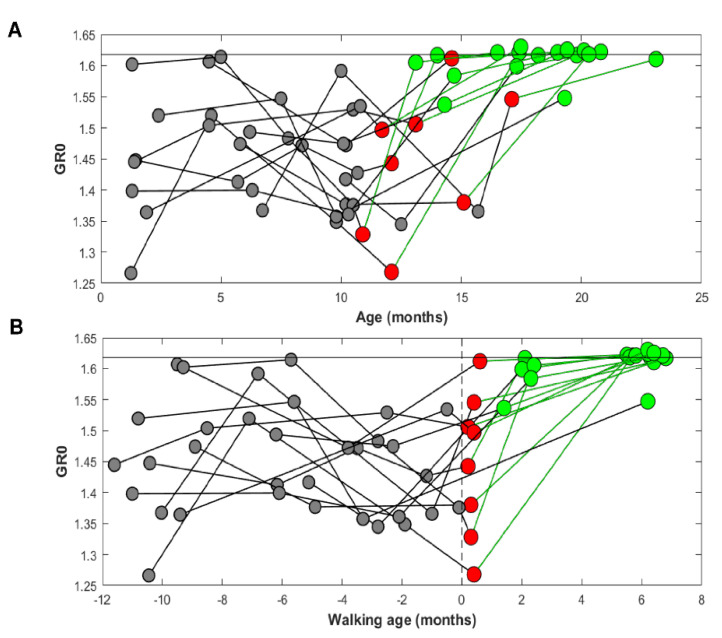
Gait ratio GR0 as a function of age (panel **A**) and walking age (panel **B**). Each line corresponds to the same toddler recorded in different motor developmental stages. Grey dots depict sessions of toddlers before the beginning of independent walking; red dots are sessions of toddlers at their first steps and green dots comprise sessions of toddlers after at least 1 month of walking experience up to six months of age.

**Figure 3 brainsci-12-00155-f003:**
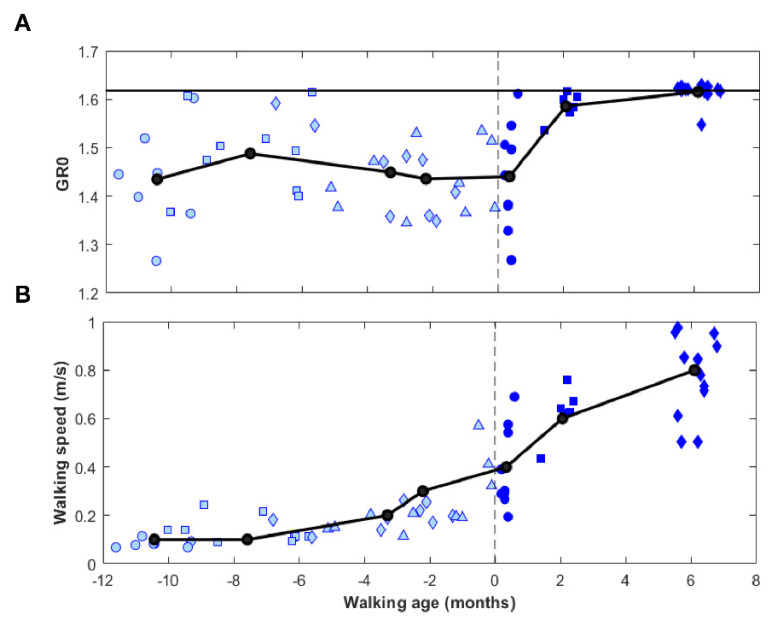
Gait parameter GR0 (panel **A**) and walking speed (panel **B**) as a function of walking age (blue dots) and relevant mean values averaged for the seven groups (1–7). Light blue color refers to the sessions before the first steps (Group 1 = circles 0–3 months, Group 2 = squares 3–7 months, Group 3 = diamonds 7–10 months, Group 4 = triangles: before first steps,). Dark blue color depicts sessions at and after the first steps (Group 5 = dark blue circles: first steps, Group 6 = dark blue squares: around two months of walking experience, Group 7 = dark blue diamonds: around six months of walking experience).

**Table 1 brainsci-12-00155-t001:** Results of the fitting applied in Figure 1. The equation of the fit is f(t) = *a* ∗ exp(−t/τ) + *b*, with R^2^ being the determination coefficient for the three gait ratios: GR0 (stride duration/stance duration), GR1 (stance duration/swing duration), and GR2 (double support duration/swing duration).

Time	Gait Ratios	*a*	τ (Months)	*b*	R^2^
Before first steps	GR0	−0.022	−7.067	1.451	0.15
	GR1	−102.9	15,762.93	105.4	0.06
	GR2	248.2	32,092.43	−247.4	0.06
From first steps and after them	GR0	−0.446	0.218	1.618	0.89
	GR1	0.841	0.882	1.629	0.90
	GR2	−0.960	1.369	1.515	0.75

**Table 2 brainsci-12-00155-t002:** Results of the correlational analysis. GR0 is operationalized as the ratio between stride and stance. “All stages” refer to all seven subgroups Group 1–Group 7. “Before first steps” includes all toddlers recorded before the walking onset (Group 1–Group 4 subgroups). “First steps and after” entails the toddlers up to six months after their independent walking onset (Group 5–Group 7 subgroups).

Pearson’s Correlations	Stages of Motor Development	Age	Walking Age	Walking Speed	Froude Number
GR0	All stages	0.487 **	0.496 **	0.694 **	0.678 **
	Before first steps	−0.162	−0.113	0.137	0.176
	First steps and after	0.660 **	0.697 **	0.852 **	0.738 **
Walking Speed	All stages	0.828 **	0.865 **	-	0.968 **
	Before first steps	0.542 **	0.662 **	-	0.944 **
	First steps and after	0.637 **	0.746 **	-	0.966 **
Froude Number	All stages	0.727 **	0.775 **	0.968 **	-
	Before first steps	0.364 *	0.492 *	0.944 **	-
	First steps and after	0.510 *	0.684 **	0.966 **	-

Significant values are marked as ** *p* < 0.001, * *p* < 0.05.

## Data Availability

The data that support the findings of this study are available from the corresponding author, upon reasonable request.

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
