# Peer review of "The Role of Walking Experience in the Emergence of Gait Harmony in Typically Developing Toddlers"

_brainsci, 2022, doi:10.3390/brainsci12020155_

Round 1

Reviewer 1 Report

Main comments

The area of studies on the mechanical determinants of toddler walking is extremely interesting as it answers an extremely important question of functional mobility and at the same time is an area of interaction between neuroscience, physiology and biomechanics. The study aimed to analyze the role of independent walking time on gait harmony in toddlers.

And, indeed, independent walking time is an important factor for basic gait timing.

My suggestions are of a secondary nature. The paper in general is well written and quite organized. I suggest inserting the study design in the abstract and methods in the main text (retrospective observational).

My only important general comment is about the mechanism that induces or explains the golden ratios in human gait, which are related to the pendulum mechanism determined by Cavagna and co-workers (e.g., Cavagna, Thys and Zamboni, J Physiol, 1976). I have missed this explanation. Otherwise, the determination of the golden ratios of human gait seem throughout the text, somewhat haphazard, whereas these temporal patterns are directly related to the fundamental minimizing mechanism of human gait, the inverted pendulum mechanism.

Minor points

Line 157 – Here, more than confirming, you analyzed indeed by visual inspection the video registration.

Some details of the data analysis procedures are missing:

- how strictly did the two raters of the gait events work? (was there a third party for when the results were not similar? If there was, what was the acceptable degree of difference?).

- Why were only two markers used and what was the function? Were algorithms used to determine the events?

- The position data for these anatomical markers were used for what?

Appendix A – kg, then, body mass (not weight). Still, consider using two decimal places in the column of speed

Lines 317-318 – consider including child obesity as probable focus to future applications of these ratios (PMID: 32441842).

Reviewer 2 Report

The introduction is well organized; however, the rationale is weak and would gain to be consolidated. Generally, the sentences which are related to the gold ratio are too much peremptory: one idea presented as a fact and few or no argumentation nor citation.

More rationale must be given. Honestly, give me any irrational number and I will find a relationship with a gait feature. My stance phase is 59.7% and my double support is 19%, 59.7/19 = 3.14. Maybe π is a magic gait number? I voluntarily enlarge the line, not for pure criticism, but I consider this part can be highly better.

I agree with the gait cycle to stance ratio and the stance to swing support ratio. But you need to explain about the swing to double support ratio. Looking under different forms of double support (one double support or both double supports considered), I do not see the Phi value.

Regarding fractality, let’s assume we can use the term fractal for the stance as a portion of the gait cycle, then the single support (equivalent to the contralateral swing) as a portion of the stance, but I disagree with the consideration of the double support as a fractal element of the single support. Looking at different gait features, such as the gold ratio, I can surely see what seems to be fractals somewhere.

Harmony of walking must be defined.

Why do you think that studying “stepping” before the onset of walking would be of interest? Like it is now, it looks like a research team publishing about the same concept of gold ratio and trying to apply it to any data/domain.

Do you consider what is recorded before the onset of walking can be compared with what is recorded after the onset? There is also a lot of confounding effects due to the treadmill, to the manner the parents are supporting the child.

There is an issue with the number of strides computed and considered. You should have defined rules to avoid comparing a performance based on 3 strides with another with 24 or 600. Using only the mean value for each parameter is restrictive and could bias the results. Please, look at the confidence intervals and provide intra-variability of the gait ratio and inter-children variability. Based on the wide range of intra- and inter-variability seen in toddlers, and even in older children, the current results don’t provide veritable implication for the knowledge in motor learning or for clinical usage.

But again, the rational is not strong enough in the current version, what is impacting the whole quality of this work.

English must be seen by a native English speaker, there is a lot of errors. 

Round 2

Reviewer 2 Report

It was a pleasure to read this updated version, because a real improvement has been done throughout the manuscript.